# Exploration and Comparison of the Effect of Conventional and Advanced Modeling Algorithms on Landslide Susceptibility Prediction: A Case Study from Yadong Country, Tibet

**Zhu Liang** [1,2,3,4], **Weiping Peng** [1,2,3], **Wei Liu** [1,2,3], **Houzan Huang** [1,2,3], **Jiaming Huang** [1,2,3], **Kangming Lou** [1,2,3], **Guochao Liu** [1,2,3] and **Kaihua Jiang** [1,2,3,*]

1   Guangzhou Urban Planning & Design Survey Research Institute, Guangzhou 510060, China; liangzhu19@mails.jlu.edu.cn (Z.L.); pengweiping@gzpi.com.cn (W.P.); liuwei@gzpi.com.cn (W.L.); huanghouzan@foxmail.com (H.H.); 13632355053@163.com (J.H.); loukangming@126.com (K.L.)
2   Guangdong Enterprise Key Laboratory for Urban Sensing, Monitoring and Early Warning, Guangzhou 510060, China
3   Guangzhou Collaborative Innovation Center of Natural Resources Planning and Marine Technology, Guangzhou 510060, China
4   College of Construction Engineering, Jilin University, Changchun 130012, China
*   Correspondence: ytgcs@gzpi.com.cn

**Abstract:** Shallow landslides pose serious threats to human existence and economic development, especially in the Himalayan areas. Landslide susceptibility mapping (LSM) is a proven way for minimizing the hazard and risk of landslides. Modeling as an essential step, various algorithms have been applied to LSM, but no consensus exists on which model is most suitable or best. In this study, information value (IV) and logistic regression (LR) were selected as representatives of the conventional algorithms, categorical boosting (CatBoost), and conventional neural networks (CNN) as the advanced algorithms, for LSM in Yadong County, and their performance was compared. To begin with, 496 historical landslide events were compiled into a landslide inventory map, followed by a list of 11 conditioning factors, forming a data set. Secondly, the data set was randomly divided into two parts, 80% of which was used for modeling and 20% for validation. Finally, the area under the curve (AUC) and statistical metrics were applied to validate and compare the performance of the models. The results showed that the CNN model performed the best (sensitivity = 79.38%, specificity = 91.00%, accuracy = 85.28%, and AUC = 0.908), while the LR model performed the worst (sensitivity = 79.38%, specificity = 76.00%, accuracy = 77.66%, and AUC = 0.838) and the CatBoost model performed better (sensitivity = 76.28%, specificity = 85.00%, accuracy = 80.81%, and AUC = 0.893). Moreover, the LSM constructed by the CNN model did a more reasonable prediction of the distribution of susceptible areas. As for feature selection, a more detailed analysis of conditioning factors was conducted, but the results were uncertain. The result analyzed by GI may be more reliable but fluctuates with the amount of data. The conclusion reveals that the accuracy of LSM can be further improved with the advancement of algorithms, by determining more representative features, which serve as a more effective guide for land use planning in the study area or other highlands where landslides are frequent.

**Keywords:** landslide susceptibility; information value; logistic regression; machine learning; deep learning; GIS

## 1. Introduction

Landslides are a geological phenomenon that can cause extensive property damage and pose a threat to the safety of residents worldwide [1–3]. Notably, China experiences a higher frequency and larger scale of landslides than any other country [4,5]. Effective prevention measures require the identification and mapping of existing landslides, with a focus

on predicting potential disaster areas [6,7]. To minimize the hazard and risk of landslides, landslide susceptibility mapping (LSM) is widely recognized as a proven approach.

The precision of LSM is contingent upon dependable data and modeling algorithms. Throughout time, a range of algorithms have been utilized in LSM, allowing for incremental improvements in accuracy. Both quantitative (data-driven) and qualitative (knowledge-based or physically based) algorithms are viable options for landslide susceptibility modeling [8,9]. Data-driven algorithms are usually classified as bivariate methods (such as certainty factor (CF) and information value (IV)), multivariate methods (such as logistic regression (LR) and cluster analysis), conventional machine learning [10–13], and deep learning [14]. Notwithstanding the fact that knowledge-based methods tend to be subjective and confined to small-scale domains, physically based methods, which rely on physical experiments, often entail significant time and financial resources and exhibit limited scalability. In recent years, the advent of advanced remote sensing has facilitated the acquisition of landslide samples and critical factors, rendering data-driven approaches more accessible and precise. Statistical methods are sensitive to the distribution and collinearity of data, and certain assumptions must be satisfied before application. There have been a number of conventional machine learning methods (such as random forest and categorical boosting (CatBoost) observed due to their ability to solve non-linear geo-environmental problems without making unnecessary assumptions [15,16]. The structure of conventional machine learning methods is relatively simple, usually involving only 1~2 hidden layers. While certain studies have undertaken a thorough examination and comparison of conventional statistical and machine learning models, a consensus has yet to be reached regarding the optimal or most appropriate model [17,18].

Deep learning represents a progression beyond machine learning, exhibiting superior performance in identifying representative features and potentially enhancing classification accuracy [19–21]. Convolutional neural networks (CNN), a prominent example of deep learning, have been extensively validated in domains such as image classification and facial recognition, yet their application to LSM remains limited [22–24]. A multi-level structure of CNN is constructed to explore the complex non-linear relationships between variables, which is accorded with the characteristics of LSM.

The superiority of the study is to assess the impact of algorithms on prediction accuracy and feature recognition by comparing the performance of conventional and advanced algorithms, as well as four representative models (IV, LR, CatBoost, and CNN). Yadong County in Southeastern Tibet was selected as the study area because of its topographic and geological conditions, resulting in frequent shallow landslides. Four models, IV, LR, CatBoost, and CNN, were explored and compared for the effect on landslide susceptibility prediction.

## 2. Materials

### 2.1. Study Area

Yadong County, situated at the southern foot of the middle section of the Himalayas, falls under the jurisdiction of Shigatse City, Tibet Autonomous Region (Figure 1). The study area, spanning over 4240 km$^2$, is home to a population of over 120,000 individuals. The region is characterized by a high mountain landform, with an average elevation of 3500 m (ranging from 1747~7057 m). The study area is geographically divided into two distinct parts, namely north and south, based on the line from Pali Town to Kangbu Town, owing to the significant difference in altitude. Consequently, two distinct climates have emerged in these regions [5]. The study area exhibits distinct geographical and geological characteristics. The northern region is situated at an elevation of 4300 m above sea level and experiences a cold and dry climate, with an average annual rainfall of approximately 410 mm. Conversely, the southern region has a lower average altitude of 2800 m, a humid climate, and an annual precipitation of 873 mm. The geological composition of the study area is primarily composed of shale, limestone, and dolomite, with the southern region being characterized by faults and folds. There is a degree of VIII on the modified Mercalli index, indicating a high seismic intensity. The southern area also exhibits a higher density of

geological disasters, such as collapses and landslides, while the northern region is primarily affected by debris flows. Figure 1 also shows the location of landslides that occurred over the years and roads distributed in the study areas.

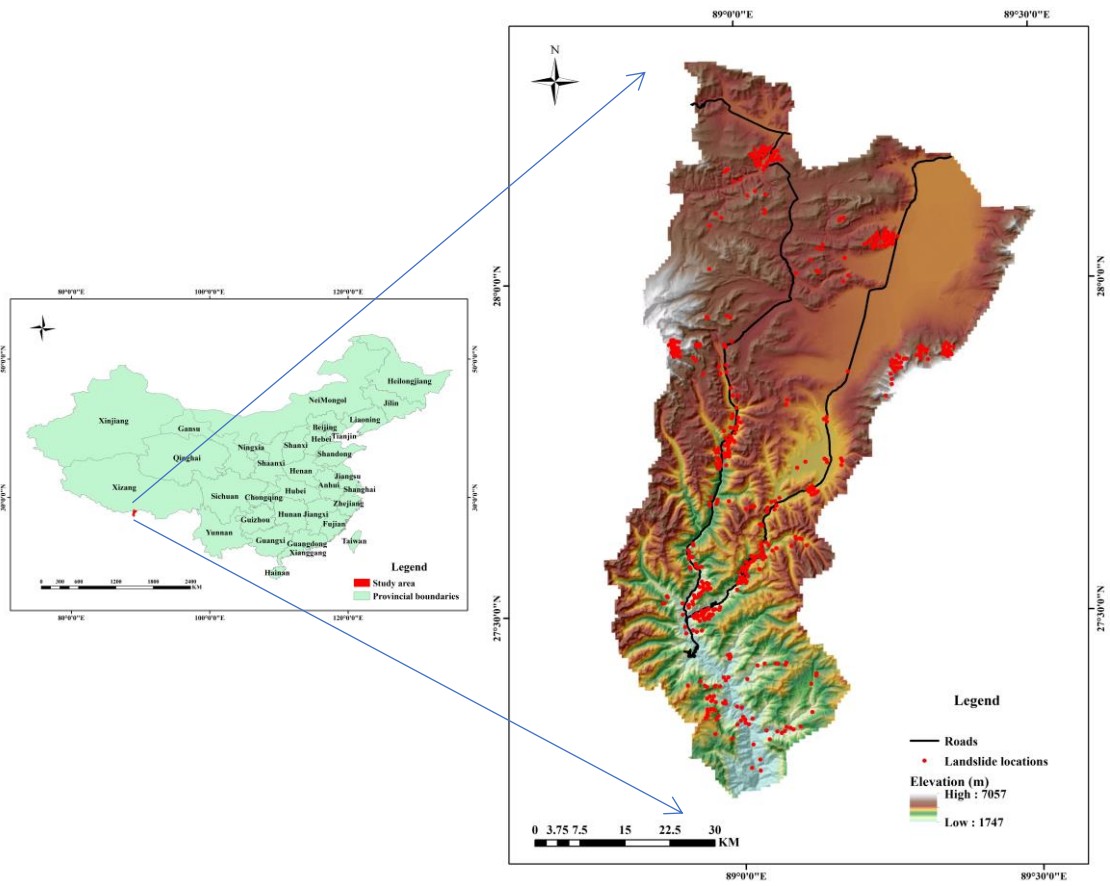

**Figure 1.** Location of the study area showing elevation and landslide samples.

On 18 September 2011, a magnitude 6.8 earthquake occurred in Sikkim, India. The study area was affected by the earthquake, which induced a series of secondary geological disasters, causing great harm to local residents and roads (Figures 2 and 3). Therefore, it is of great significance to compile an accurate landslide susceptibility map for the study area.

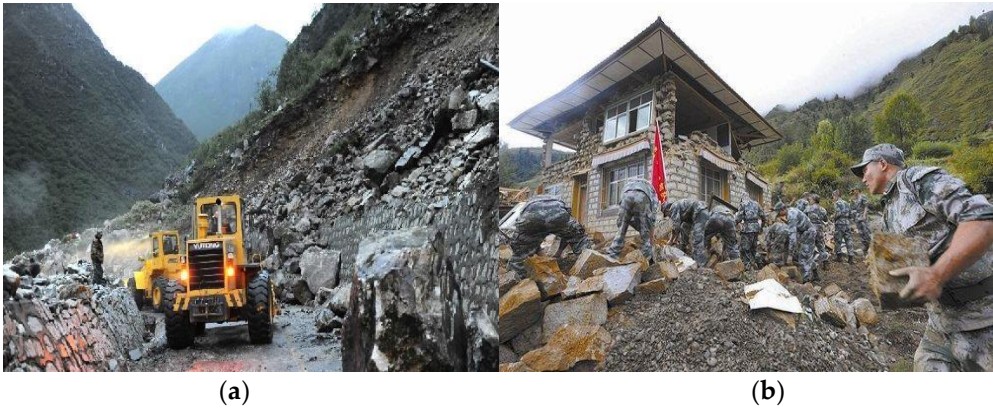

| (a) | (b) |

**Figure 2.** The Sikkim earthquake triggered landslides in Yadong County: (**a**) S204 road was interrupted; (**b**) houses collapsed.

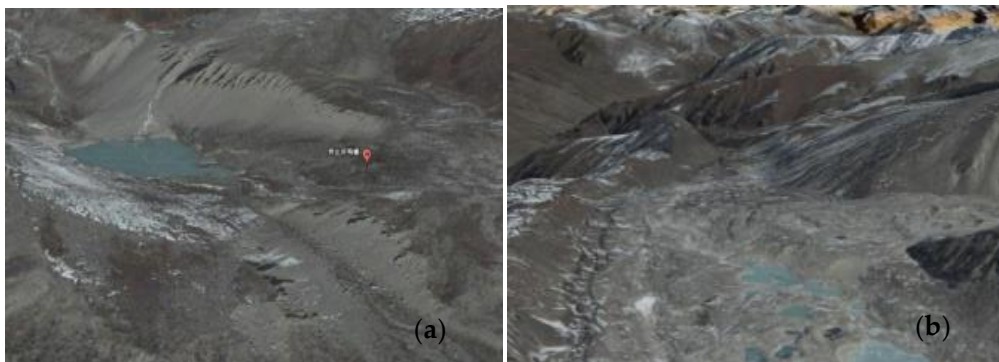

**Figure 3.** Remote sensing image interpretation: (**a**) debris flow in Kangbu township; (**b**) debris flow in Duina township.

### 2.2. Data Preparation

#### 2.2.1. Landslide Inventory

Typically, data-driven methods for LSM assume that landslides have a greater chance of recurring under the same conditions as they did before [25]. Therefore, a comprehensive and exact landslide inventory that shows the locations and numbers of landslides is essential [7]. It is therefore essential to develop a complete and accurate landslide inventory (Figure 4) that shows the locations and quantities of landslides. Landslides were obtained by aerial photographs, collecting literature and historical reports, and conducting extensive fieldwork. Landslides are bounded by polygons containing the entire perimeter, and 496 polygons representing the landslide perimeter were identified. The landslide locations are the centroids of the polygons (Figure 1). Landslides are mainly small- and medium-sized shallow landslides with volumes ranging from 100 m$^3$ to 1000 m$^3$. Using a 1:1 sampling strategy [26], the model was trained and validated with 992 samples, including 496 landslides with a sign of "1" and 496 non-landslides with a sign of "0". Non-landslide samples were randomly selected from non-landslide dense areas.

#### 2.2.2. Conditioning Factors

A total of 11 condition factors were selected based on the study region's characteristics, the data's availability, reliability, and practicality, including elevation, slope, maximum elevation difference (MED), plan curvature, profile curvature, topographic wetness index, distance to faults, distance to roads, distance to streams, annual rainfall, and NDVI [27].

Elevation, which was divided into five sub-classes by 1000 m intervals, has an influence on both rainfall and vegetation [28,29] (Figure 4a). Slope, which controls shear strength at potential slide surfaces as well as subsurface flow [30], was reclassified into six classes by 10° intervals (Figure 4b). It was calculated in ArcGIS 10.2 that the MED reflects potential kinetic energy of slope units. By 100 m intervals, the thematic map was reclassified into six classes (Figure 4c). Erosion and deposition can be determined from slope curvature since it is essential to the geometry of slopes [31]. Six classes were established for profile and plan curvatures (Figure 4d,e). The TWI represents basic terrain, which was divided into six categories (Figure 4f). With the spatial resolution of 30 m, six topographic factors were extracted from the digital elevation model (DEM). The DEM was originally sourced from the Geospatial Data Cloud, which was accessed on 6 January 2023.

Bulk-rock strength could be reduced as a result of faults acting as potential weak planes in slopes, and distance to fault was constructed with six classes by 2000 m intervals (Figure 4g). In a similar manner, distances to the road and river were divided into six categories with an interval of 2000 m (Figure 4h,i).

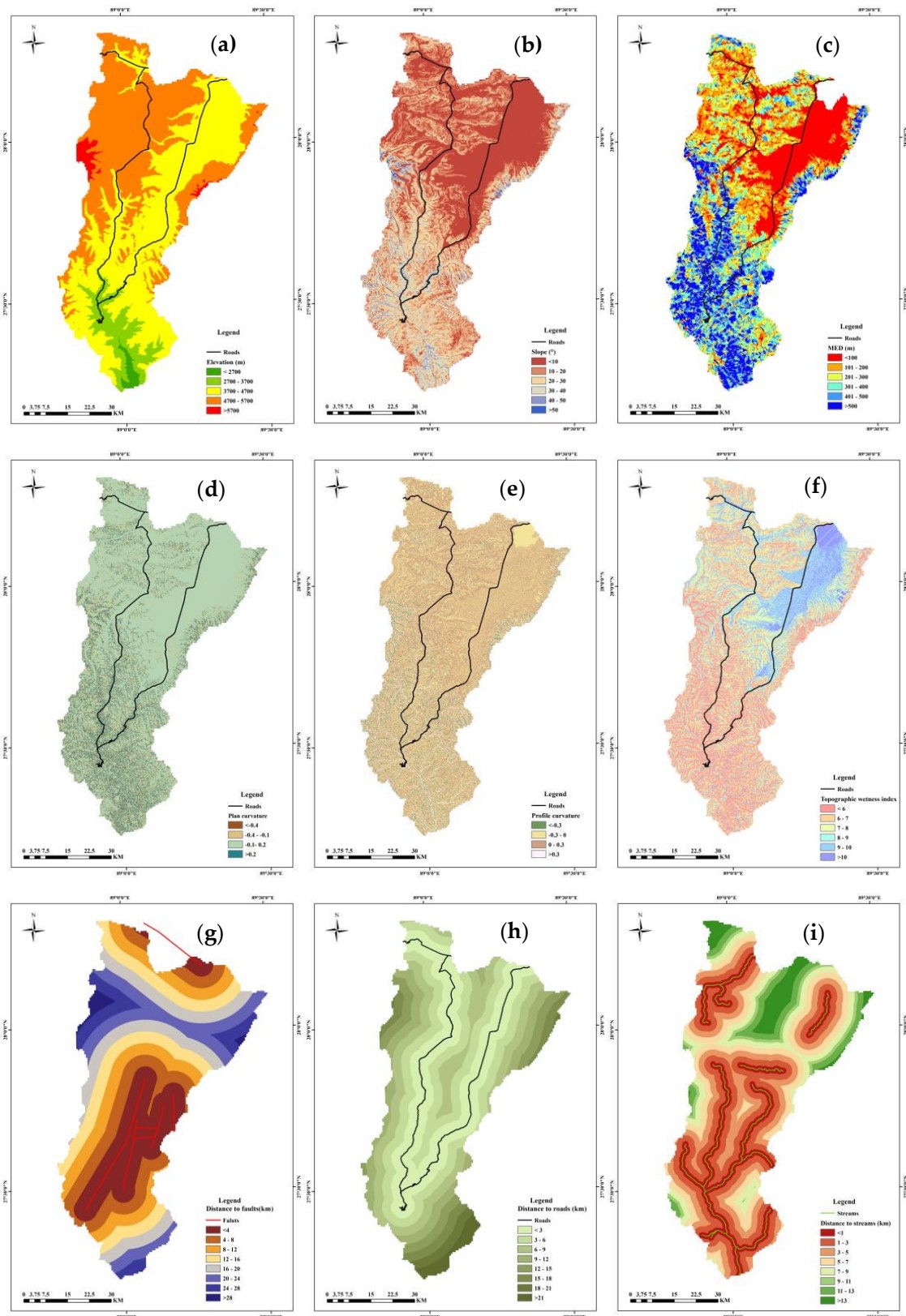

**Figure 4.** *Cont.*

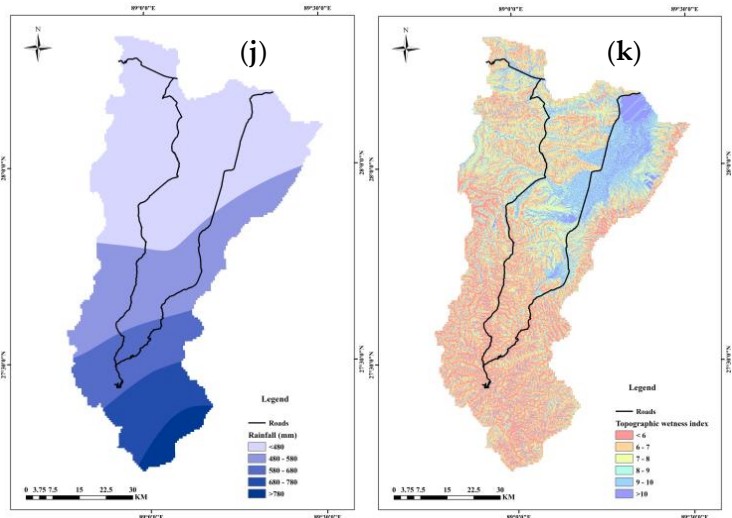

**Figure 4.** Study area thematic maps: (**a**) elevation; (**b**) slope; (**c**) MED; (**d**) plan curvature; (**e**) profile curvature; (**f**) TWI; (**g**) DTF; (**h**) DTR; (**i**) DTS; (**j**) rainfall; (**k**) NDVI.

An existing 1:50,000 geological map was used to extract fault information, and Landsat 8 LOI images were used to determine road and river network information. Using the Euclidean Distance ArcGIS Tool, the distances from each raster unit of the area to the nearest fault, road, stream, and highway were calculated.

The unique trigger factor considered in the study was rainfall, and it has been applied numerous times. Based on the data from 14 precipitation stations near the study area, an ordinary kriging interpolation was used to generate the thematic map in ArcGIS. The thematic map was reclassified into 4 classes (Figure 4j).

The consolidation of vegetation roots can stabilize soil, which alleviates the effect of rainfall on the stability of the slope [32]. The vegetation normalization index (*NDVI*) was applied to evaluate the vegetation, and it is calculated by:

$$NDVI = \frac{NIR - RED}{NIR + RED} \tag{1}$$

where *RED* is the reflection value of the red wavelength and *NIR* is the near-infrared wavelength.

*NDVI* ranges from −1~1 with a positive value indicating coverage, and the greater the value, the greater the coverage. The thematic map was reclassified into 6 classes (Figure 4k). The *NDVI* was derived from Landsat 4–5 TM satellite images.

### 2.2.3. Mapping Units

LSM commonly uses grid cells, slope units, and unique-condition units as mapping units. The choice of mapping unit is controversial; however, grid cells are the most common [33,34]. Landslides can be represented better by a slope unit, which represents their source, transport, and accumulation areas [35–37]. Thus, the study area is divided into 25,483 slope units with the hydrologic analysis tool in ArcGIS and necessary artificial corrections are accompanied according to remote sensing images.

### 3. Methods

The main aims of the study are to explore and compare the effect of conventional and advanced modeling algorithms on landslide susceptibility prediction. The methodology followed in this study mainly contains four steps (Figure 5). Firstly, data sets including landslide samples, conditioning factors, and mapping units are prepared for modeling. The second step is to divide the data set into training and validation parts. After that, four representative algorithms, IV, LR, CatBoost, and CNN, are applied to LSM. Finally, the

performance of LR, CatBoost, and CNN models are analyzed and compared based on some key measures.

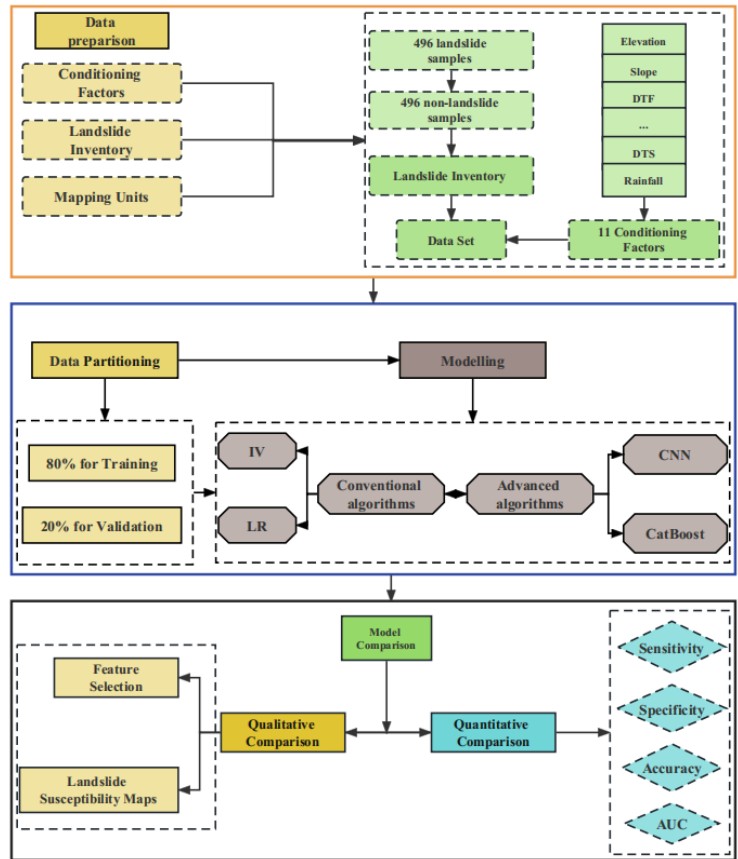

**Figure 5.** Flow chart of this study.

### 3.1. IV

By using the frequency or density of landslides, the IV method reflects the magnitude of the hazards associated with different influencing factors and their sub-intervals. IV was first proposed by Yin and Yan (1988) and later modified by Van Westen (1993) [38]. Equation (1) shows the method for the calculation of the information values:

$$I(A_{i-j}) = \ln \frac{s_{i-j}/S}{n_{i-j}/N} \tag{2}$$

where $i = 1, 2, 3, \ldots, n$; $j = 1, 2, 3, \ldots, m$; $s_{i-j}$ represents the area of the landslide of the $i$-th conditioning factor in $j$-th interval; $n_{i-j}$ represents the area of the $i$-th conditioning factor in $j$-th interval; $S$ represents the total area of the landslide; $N$ represents the total area.

Depending on the IV method, the value might be positive or negative. When IV values are positive, they indicate that the factor is conducive to the occurrence of landslides in a particular interval; a higher IV indicates a higher likelihood of landslides, and vice versa.

The total information value I can be determined by:

$$I_i = \sum_{i=1}^{n} \ln \frac{s_i/S}{n_i/N} \tag{3}$$

Finally, the information value calculated by each unit is processed by linear normalization.

### 3.2. LR

The LR model is used for statistical analysis of binary dependent variables (the dependent variable y has two values: 0 and 1) [39]. The LR model is advantageous since the data distribution can either be nominal or continuous [40]. It is computed using the following equation:

$$P = \frac{1}{1 + e^{-y}} \tag{4}$$

where $P$ represents the occurrence probability of an event (such as a landslide) on the s-curve in the range of 0 to 1; y represents a linear combination function and is calculated by using Equation (5).

$$y = b_0 + b_1 x_1 + b_2 x_2 + b_3 x_3 + b_n x_n \tag{5}$$

where $b_0$ is the constant value or intercept of the equation, and $b_1, b_2, ..., b_n$ are the regression coefficients of the explanatory variables $x_1, x_2, ..., x_n$.

In the current study, the LR model was performed in SPSS software, and the forward stepwise method was adopted to screen valuation indexes. The conditioning factors were calculated as independent variables, whereas dependent variables represent the occurrence of landslides. During the last step of the analysis, all variables were significant at less than 0.05, so no additional variables were included.

### 3.3. CatBoost

CatBoost is an improved implementation based on the Gradient Boosted Decision Tree (GBDT) framework, with fewer parameters. It is an open-source machine-learning library of Yandex and a member of the Boosting family. The gradient deviation and prediction offset can be effectively solved by the CatBoost algorithm, so as to reduce the occurrence of over-fitting and improve the accuracy and generalization ability of the algorithm [41]. It is worth noting that the selection of different random numbers can have an impact on the predictive outcome of the model. Detailed information about CatBoost can be found in other literature [42].

### 3.4. CNN

CNN provides an end-to-end learning model that can select model parameters through traditional gradient descent methods. The trained CNN can learn image features and complete the extraction and classification of image features [43]. A CNN consists of the network structures as convolutional layer, down-sampling layer, and fully connected layer, each of which contains a number of independent neurons [44]. Its notable feature is that the weight sharing and local connection of the convolution kernel in the hidden layer can greatly reduce the number of weights, thereby reducing the complexity of the convolutional network model.

The unique convolutional layer and pooling layer structure of CNN can be combined arbitrarily to obtain an infinite variety of network models. This study applied the AlexNet model (including 4 convolutional layers and 3 fully connected layers) to LSM [45]. Before modeling, conditioning factors are requested to be superimposed to obtain a multi-band two-dimensional image as the input raw data in LSM. Since LSM belongs to a two-classification problem of landslide disasters, two neurons are placed in the output layer classifier to represent landslide and non-landslide. All parameters have been optimized through a trial-and-error approach. The number of training cycles is 10, the initial learning rate is 0.01, the loss function uses standard cross-entropy, the optimizer is Adam, and the activation function is Relu.

The machine learning and deep learning-based algorithms are implemented in Python3.7 based on the Package of Numpy, Scikit-Learn, and Tensorflow. The statistics-based algorithms are implemented in SPSS.

### 3.5. Models Evaluation

A predictive model will not be convincing without scientific validation, so existing data will need to be split into training and validation sets. Previous research derived performance measures for single-hold-out models by employing a single training set and an independent test set. In the study, 80% of the data sets were randomly chosen for training and 20% for validation in LSM, and the split was repeated 5 times. Finally, these models were evaluated using the average of 5 groups.

Four evaluation measures—accuracy, sensitivity, specificity, and AUC—were combined to analyze the performance of the models. Accuracy, sensitivity, and specificity are calculated from the Confusion Matrix, which is an N × N matrix (Table 1). The TP represents the number of landslides that have been correctly predicted as unstable, and TN represents the number of non-landslides that have been correctly predicted as stable, while FP represents the number of non-landslides that have been predicted incorrectly as unstable and FN represents the number of landslide units that have been predicted incorrectly as stable.

**Table 1.** Confusion matrix analysis with evaluation measures.

| | | Actual Values | | Accuracy | Sensitivity | Specificity |
|---|---|---|---|---|---|---|
| | | Positive (1) | Negative (0) | | | |
| Predicted Values | Positive (1) | True Positive (TP) | False Negative (FN) | (TP + TN)/ (TP + TN + FP + FN) | TN/(TN + FP) | TN/(TN + FP) |
| | Negative (0) | False Positive (FP) | True Negative (TN) | | | |

An additional indicator of model validity is the area under the receiver operating characteristic curve (AUROC). The value of AUROC ranges from 0.5~1; the larger the value, the better the generalization ability of the model and prediction performance. Finally, the natural breaks method was applied to reclassify landslide susceptibility into five classes: very low, low, moderate, high, and very high.

## 4. Results

### 4.1. Performance and Comparison of Conventional and Advanced Algorithms

Before modeling, the data were normalized with Z-scores to eliminate the effects of different dimensions. Additionally, a correlation analysis was conducted to test the collinearity among the independent variables using the variance inflation factor (VIF) [44]. There is severe collinearity between the selected variables if the VIF value exceeds 10. Table 2 showed the VIF values of the chosen independent variables, and the factor elevation has the highest VIF value (5.711). The result indicates that no severe collinearity problems exist among the chosen variables, and thus 11 conditioning factors were applied to the modeling.

The performance of the three models using the confusion matrix is shown in Table 3 As for training, the CNN model achieved the highest value of sensitivity (84.21%), followed by the CatBoost model (sensitivity = 82.96%) and the LR model (sensitivity = 81.45%). As for specificity, the CNN model performed best with 93.52%, followed by the CatBoost model at 86.63% and the LR model at 84.79%. The CNN model had the best accuracy and ROC values of 88.88% and 0.944, while the LR model had the worst values of 83.13% and 0.897. As well, CatBoost performed well, scoring 86.63% and 0.930, respectively.

**Table 2.** Multicollinearity diagnosis indexes for variables.

| Variables | VIF |
|---|---|
| Elevation | 5.117 |
| Slope | 3.426 |
| MED | 5.726 |
| Plan curvature | 1.499 |
| Profile curvature | 1.291 |
| TWI | 6.071 |
| Distance to fault | 2.641 |
| Distance to stream | 4.492 |
| Distance to road | 4.302 |
| Average annual precipitation | 1.763 |
| NDVI | 2.697 |

**Table 3.** Models' performance using related indices.

| Parameter | Training | | | Validation | | |
|---|---|---|---|---|---|---|
| | LR | CatBoots | CNN | LR | CatBoots | CNN |
| Sensitivity (%) | 81.45 | 82.96 | 84.21 | 79.38 | 76.28 | 79.38 |
| Specificity (%) | 84.79 | 90.27 | 93.52 | 76.00 | 85.00 | 91.00 |
| Accuracy (%) | 83.13 | 86.63 | 88.88 | 77.66 | 80.71 | 85.28 |
| AUC | 0.897 | 0.930 | 0.944 | 0.838 | 0.893 | 0.908 |

The verification data set is more useful and important for evaluating the ability of these models to generalize. In Figure 6, we found that the CNN model had the highest sensitivity, specificity, accuracy, and AUC values, namely 79.38%, 91.00%, 85.28%, and 0.908. Additionally, CatBoost performed well with 76.28%, 85.00%, 80.71%, and 0.893, respectively. The LR model remained the worst with values of 79.38%, 76.00%, 77.66%, and 0.838 (Table 3).

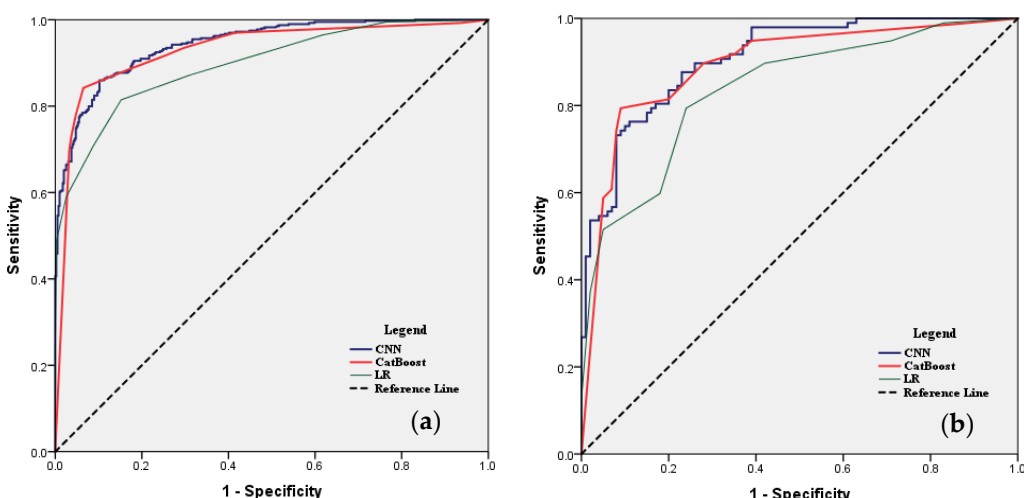

**Figure 6.** Analysis of ROC curve for the landslide susceptibility map: (**a**) success rate curve of landslide using the training data set; (**b**) prediction rate curve of landslide using the validation dataset.

Model performance declined in verification, particularly for the LR and CatBoost models (the accuracy value decreased by about 6%), which indicated that the models were overfitting and generalizability was suspicious. Meanwhile, the performance of the CNN model was stable as the value of AUC reached 0.908, and the accuracy value was close to the training data set.

## 4.2. Evaluation of Conditioning Factors

### 4.2.1. Application of Conventional Algorithms

Bivariate methods were used to establish relationships between the conditioning factors and the occurrence of the landslides. From Table 4, as for elevation, the percentages of landslide area for 3700~4700 m and 4700~5700 m were 43.80% and 28.63%, respectively, which means that over 70% of landslide areas were distributed among the two classes but the IV value of these two classes were both negative, while the highest IV of elevation is 1.098 for the class of >5700. For slope, the IV values increased as the slope increased, as well as the factor MED, which indicated that landslides are more likely to occur in steep areas. The IV of slope ranged from −1.106~1.709, in which the classes of >20 were all positive. In terms of TWI, the class of 6~7 had the highest IV of 0.371 and the IV were all negative at the class of >8.

**Table 4.** Landslide conditioning factors in this study.

| Conditioning Factors | Zone | Ni/N | Si/S | IV |
|---|---|---|---|---|
| Elevation (m) | <2700 | 1.21% | 1.09% | 0.100 |
| | 2700~3700 | 21.98% | 8.26% | 0.978 |
| | 3700~4700 | 43.80% | 49.88% | −0.131 |
| | 4700~5700 | 28.63% | 39.29% | −0.317 |
| | >5700 | 4.44% | 1.48% | 1.098 |
| Slope (°) | <10 | 11.90% | 35.95% | −1.106 |
| | 10~20 | 25.60% | 29.88% | −0.154 |
| | 20~30 | 38.31% | 25.67% | 0.400 |
| | 30~40 | 22.18% | 8.13% | 1.003 |
| | >40 | 2.02% | 0.36% | 1.709 |
| MED (m) | <100 | 7.46% | 36.41% | −1.585 |
| | 100~200 | 13.91% | 21.80% | −0.449 |
| | 200~300 | 11.90% | 15.82% | −0.285 |
| | 300~400 | 20.56% | 10.76% | 0.648 |
| | >400 | 46.17% | 15.21% | 1.110 |
| Plan curvature | <−0.4 | 0.20% | 0.12% | 0.538 |
| | −0.4~−0.1 | 4.44% | 4.89% | −0.097 |
| | −0.1~0.2 | 92.74% | 93.11% | −0.004 |
| | >0.2 | 2.62% | 1.89% | 0.328 |
| Profile curvature | <−0.3 | 0.20% | 0.59% | −1.078 |
| | −0.3~0 | 31.20% | 36.57% | −0.157 |
| | 0~0.3 | 67.54% | 62.50% | 0.077 |
| | >0.3 | 1.01% | 0.33% | 1.106 |
| TWI | <6 | 7.26% | 5.85% | 0.215 |
| | 6~7 | 46.57% | 32.13% | 0.371 |
| | 7~8 | 33.27% | 27.54% | 0.189 |
| | 8~9 | 8.87% | 14.48% | −0.490 |
| | 9~10 | 2.42% | 9.11% | −1.326 |
| | >10 | 1.61% | 10.89% | −1.910 |
| Distance to faults (km) | <4 | 31.05% | 20.27% | 0.426 |
| | 4~8 | 16.33% | 14.71% | 0.105 |
| | 8~12 | 7.46% | 14.57% | −0.670 |
| | 12~16 | 22.18% | 16.00% | 0.327 |
| | 16~20 | 14.31% | 14.99% | −0.046 |
| | 20~24 | 8.67% | 12.22% | −0.343 |
| | 24~28 | 0.60% | 6.10% | −2.311 |
| | >28 | 0.20% | 1.14% | −1.734 |

**Table 4.** *Cont.*

| Conditioning Factors | Zone | Ni/N | Si/S | IV |
|---|---|---|---|---|
| Distance to streams (km) | <1 | 47.58% | 14.51% | 1.187 |
| | 1~3 | 19.35% | 25.62% | −0.280 |
| | 3~5 | 7.06% | 19.97% | −1.040 |
| | 5~7 | 6.65% | 14.66% | −0.790 |
| | 7~9 | 9.07% | 11.07% | −0.199 |
| | 9~11 | 5.44% | 7.00% | −0.252 |
| | 11~13 | 3.63% | 4.38% | −0.189 |
| | >13 | 1.61% | 2.78% | −0.544 |
| Distance to roads (km) | <3 | 57.66% | 32.20% | 0.583 |
| | 3~6 | 11.69% | 25.23% | −0.769 |
| | 6~9 | 14.11% | 18.23% | −0.256 |
| | 9~12 | 6.20% | 9.08% | −0.374 |
| | 12~15 | 5.44% | 6.37% | −0.158 |
| | 15~18 | 1.41% | 4.46% | −1.151 |
| | 18~21 | 2.62% | 2.07% | 0.237 |
| | >21 | 0.81% | 2.35% | −1.070 |
| Average annual precipitation (mm) | <480 | 32.26% | 50.99% | −0.458 |
| | 480~580 | 29.44% | 24.55% | 0.182 |
| | 580~680 | 24.60% | 11.55% | 0.756 |
| | 680~780 | 3.02% | 9.38% | −1.132 |
| | >780 | 10.69% | 3.53% | 1.107 |
| NDVI | <0.15 | 16.53% | 12.68% | 0.265 |
| | 0.15~0.3 | 25.20% | 36.62% | −0.374 |
| | 0.3~0.45 | 15.12% | 19.74% | −0.266 |
| | 0.45~0.6 | 15.93% | 17.94% | −0.119 |
| | 0.6~0.75 | 22.18% | 9.93% | 0.804 |
| | >0.75 | 5.04% | 3.09% | 0.490 |

For curvature, the highest IV of the plan curvature was 0.538, located at the class of <−0.4, and 1.106 of profile curvature at the class of >0.3. It was found that landslides were concentrated in convex terrain. On the other hand, landslides are highly concentrated near the faults, streams, and roads as the IV reached the maximum value at first class. It indicated that the development of faults, streams, and human engineering activities was conducive to the occurrence of landslides.

As for rainfall, the highest probability of landslides occurring appeared in the class of >780 mm (IV reached 1.107). Meanwhile, the IV did not increase with increasing rainfall and it indicated that landslide occurring is complex. In the case of NDVI, the class of 0.6~0.75 has the highest IV of 0.804. The IV changed erratically for NDVI although high vegetation cover helped enhance the stability of the landslide.

As for the LR model, the final regression equation is as follows:

$$y = (1.756\ MED) + (0.395\ profile.curvature) + (1.799\text{rainfall}) + (-1.641\ DTR) + (0.684\ DTS) + 0.28 \tag{6}$$

It can be found that MED, rainfall, and DTR were essential to landslide occurring as the coefficients were relatively large. While profile curvature and DTS were considered as secondary factors. The coefficient of DTR was negative, which indicated a negative effect on landslide occurring.

Conventional algorithms such as IV and LR methods were used to establish the relationships between the conditioning factors and the occurrence of the landslides in this study. In summary, the conditioning factors have an effect on landslides occurring and the impact of different factors on landslides varies in different intervals.

4.2.2. Application of Advanced Algorithms

Actually, conditioning factors have different influences on the occurrence of landslides while the bivariate methods fail to recognize the difference. Gini importance (GI) is defined as the total reduction in average nodal impurities across all trees. GI was also applied to analyze the relative importance of different factors, exploring the factor's contribution to landslide occurring.

The bigger the GI express greater the importance of factors of landslides occurring [44]. Table 5 showed the rank of the main conditioning factors of this study. The result showed that DTF was the most important factor responsible for landslides as the GI reached 5.36. Moreover, the factor MED, DTS, elevation, plan curvature, and slope were also pivotal since the GI were all greater than zero. While the other factors as TWI, rainfall, profile curvature, NDVI, and DTR showed little contribution to landslides in the study area (Figure 7).

**Table 5.** Conditioning factors assigned by the Gini index.

| Method | DTF | MED | DTS | Elevation | Plan Curvature | Slope |
|---|---|---|---|---|---|---|
| Gini index | 5.36 | 5.00 | 4.80 | 4.10 | 3.69 | 2.80 |

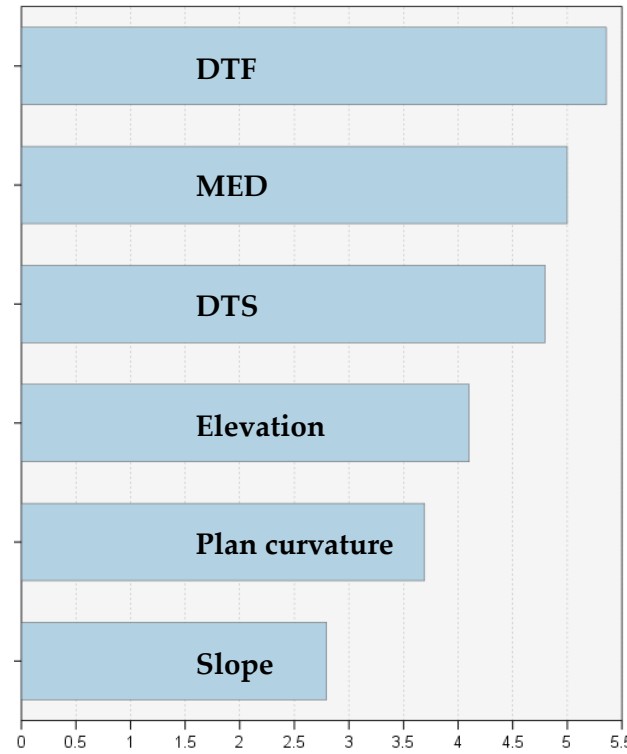

**Figure 7.** Parametric importance graphics designed by Gini index.

*4.3. Landslide Susceptibility Mapping Results*

In the case of LR model, LSI ranges from 0.11 to 0.96, and the corresponding area percentages were 40.17% (very low), 12.24% (low), 5.54% (moderate), 17.75% (high), and 24.30% (very high), respectively. Similarly, five reclassified classes of the CatBoost model accounted for 30.16%, 23.81%, 15.21%, 26.82%, and 14.94%, respectively, of the entire area. LSM constructed by the CNN model was also divided into five classes: very low (<0.18), low (<0.42), moderate (<0.653), high (<0.82), and very high (>0.82), accounting for 28.99%, 19.65%, 14.21%, 9.19%, and 27.96% of the whole area. It is noticed that LR, CatBoost, and CNN models predicted the largest proportion of very low susceptibility while IV predicted the largest proportion of very high susceptibility.

A logical landslide susceptibility map should meet two rules: (1) the density of landslide samples should increase with the increase of susceptibility class and be mainly

located in the highest susceptibility area; (2) the landslide susceptibility map should be spatially continuous and smooth, and the very high-susceptibility class area should occupy a small proportion. Concomitant with these maps (Figure 8), the landslide samples (dark spots) were mainly located in the red areas, and the non-landslide samples (blue spots) were in the green areas. The high or very high susceptibility areas were concentrated in the south of the areas, which was consistent with the distribution of landslides (Figure 9). Thus, the maps predicted by these models were logical on the whole.

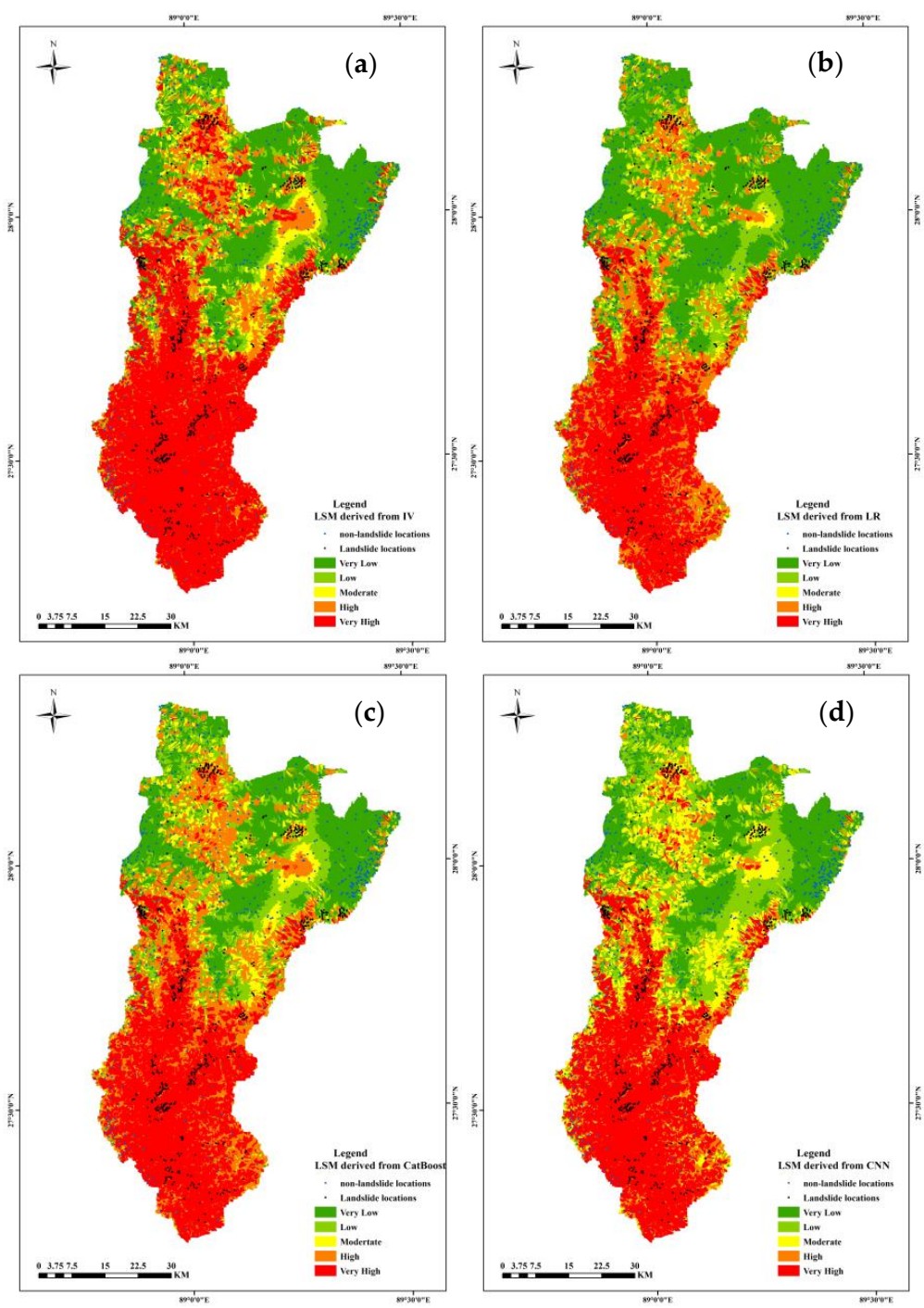

**Figure 8.** Landslide susceptibility maps: (**a**) IV; (**b**) LR model; (**c**) CatBoost model; (**d**) CNN model.

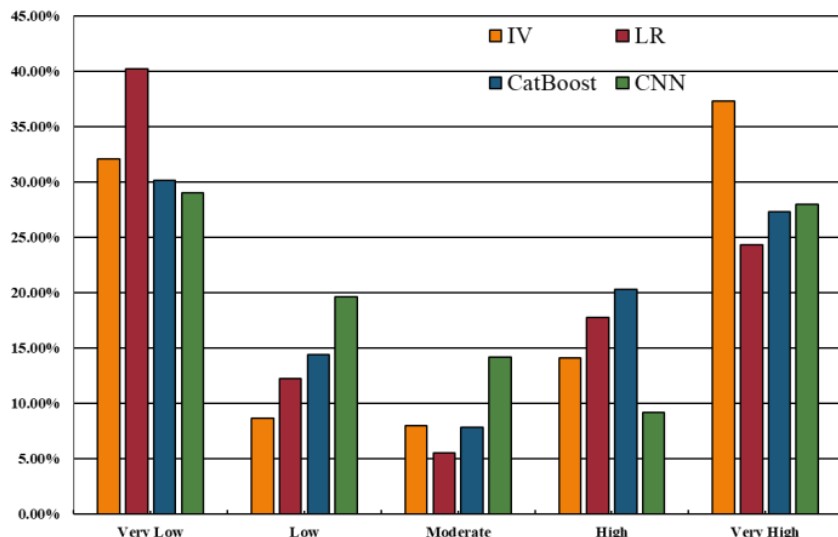

**Figure 9.** Percentages of areas in different susceptibility classes for landslide.

There are distinct differences among the landslide susceptibility maps derived from the models. Table 6 showed that the IV increased as the susceptibility class increased for the model and the IVs were all negative for the very low, low, and moderate susceptibility class while positive for the very high susceptibility class. For high susceptibility, the IVs were positive for LR, CatBoost, and CNN models while negative for the IV model. On the other hand, the maps predicted by IV and LR models were spatially discontinuous while CatBoost and CNN models produced smoother patterns. Moreover, the percentage of very high susceptibility class was similar for LR, CatBoost, and CNN models, while the IV model predicted the highest percentage, reaching 37.27%.

**Table 6.** The IV of different landslide susceptibility levels.

| Model | Class | Percentage of Area (%) | Percentage of Landslide Area (%) | IV |
|---|---|---|---|---|
| IV | Very low | 32.05% | 8.06% | −1.74 |
| | Low | 8.64% | 7.66% | −1.19 |
| | Moderate | 7.96% | 5.04% | −0.50 |
| | High | 14.08% | 20.36% | −0.14 |
| | Very high | 37.27% | 58.87% | 0.70 |
| LR | Very low | 40.17% | 5.24% | −1.60 |
| | Low | 12.24% | 4.44% | −0.47 |
| | Moderate | 5.54% | 6.05% | −0.10 |
| | High | 17.75% | 21.17% | 0.14 |
| | Very high | 24.30% | 63.10% | 0.88 |
| CatBoost | Very low | 30.16% | 5.24% | −1.74 |
| | Low | 14.42% | 4.44% | −1.18 |
| | Moderate | 7.82% | 6.05% | −0.25 |
| | High | 20.28% | 21.17% | 0.04 |
| | Very high | 27.31% | 63.10% | 0.84 |
| CNN | Very low | 28.99% | 4.44% | −1.88 |
| | Low | 19.65% | 8.67% | −0.82 |
| | Moderate | 14.21% | 12.30% | −0.14 |
| | High | 9.19% | 11.09% | 0.19 |
| | Very high | 27.96% | 63.51% | 0.69 |

## 5. Discussion

With the progress of related technologies and the maturity of theory, various approaches have been developed and applied to LSM to improve prediction accuracy and reliability [45–49]. Recently, deep learning is becoming popular and beginning to be used for LSM. In this study, CNN was applied to LSM, and its performance was compared to the conventional statistical methods and machine learning approaches.

In comparison to the other three models, the CNN model shows superior fitting and generalization capabilities in predicting landslide susceptibility. It has been found that very high landslide susceptibility areas are mainly associated with the Yarlung Zangbo River and its tributaries. A large part of the eroded slopes is scoured by the river network. The areas near streams are densely populated, and the occurrence of landslides can threaten lives.

Different algorithms have different emphases and, generally, their performance varies with different study areas [50,51]. However, advanced algorithms usually perform better in terms of accuracy compared to conventional statistical methods [52,53]. The result of our study also found that CNN and CatBoost performed better in terms of accuracy and AUC, and CNN did the best in generalization. There was a certain gap between the three models. The improvement benefits from the characteristics of the algorithms themselves for decreasing the bias, discrepancy, and over-fitting problems. It is easy to implement and acceptable in conventional statistical methods to establish a mathematical equation for investigating the relationship between factors related to landslides and landslides occurring. In machine learning methods, optimization is stressed, so the multiple parameters involved need to be tuned before application, which is unattainable for non-experts although some algorithms for optimizing parameters have been applied [54,55]. Deep learning algorithms use a more complex modeling architecture consisting of convolutional, activation, pooling, and fully connected layers, taking images as input parameters. Feature selection and information filtration are performed in the pooling layer, which is a robust step in CNN. Importantly, the dimensionality reduction is performed without changing the depth of the maps. Thus, deep learning algorithms have the ability to process data efficiently, feature extraction of high-dimensional data, and keep high prediction accuracy.

The accuracy of LSM should not be the only priority. Identifying the major conditions that lead to landslides is also critical, which assists in furthering the process [56,57]. Identifying subjective weights and objective weights allows for separating their contributions. Analytic hierarchy processes (AHP) and factor analyses (FA) are commonly used methods [58,59]. Landslides and their underlying conditioning factors can be directly correlated using LR models [60,61]. The relative importance of factors can be determined by the magnitude, plus, or minus of the coefficients. In addition, bivariate methods are capable of distinguishing factors with different susceptibilities to landslides across interval ranges. Accordingly, LR and IV are recommended to be combined to analyze the factor's contribution to a landslide occurring. GI describes the contribution of the conditioning factors by calculating the total decrease, and the final coefficients reflect the relative importance of different factors. Some studies have applied GI for feature selection, which helps to decrease redundant information [62,63].

As for bivariate methods such as IV, FR, and CF, the performance is mainly up to whether the division of the conditioning factors interval is reasonable. Some factors such as DTF and DTR in the study may show a logical distribution while the NDVI showed a chaotic distribution. However, there is no consensus on the size or number of intervals, although some methods such as natural breaks and equal intervals have been applied. Moreover, the total IV is a linear addition of the IV of all conditioning factors, which further amplifies the uncertainty of the final result. Thus, the susceptibility analysis of conditioning factors on different intervals by bivariate methods may unreliable, and the landslide susceptibility maps are also difficult to be verified quantitatively. In terms of LR, the establishment of the equation derives from the distribution characteristics of data and is sensitive to linear correlation. Thus, the performance is up to the data partition and will fluctuate during validation. The results of relative importance for conditioning factors may

be confused or even contrary to our experience on landslides. Machine learning or deep learning emphasizes iterative operations and repeated verification and requires a large amount of data. Landslide samples are limited in a restricted area and difficult to collect. The performance will also decline as the data decrease.

Moreover, lithology as a categorical variable is a crucial factor used by practically most of the researchers of LSM. The geological composition of the study area is primarily composed of shale, limestone, and dolomite. The lithology distribution in the study area is simple. Thus, we select the distance to faults as a more important factor.

## 6. Conclusions

In the current study, IV and LR were selected as representatives of the conventional algorithms, with CatBoost and CNN as the advanced algorithms, for LSM in Yadong county, and their performance was compared. The following conclusions can be drawn:

1. There was a certain gap between the models. Compared to conventional algorithms, advanced algorithms performed better in terms of prediction accuracy and CNN performed the best in generalization, thus it is regarded as the best model in this study.
2. The landslide susceptibility map predicted by CNN was more reasonable and the very high susceptibility areas were mainly distributed along the Yarlung Zangbo River.
3. As for feature selection, IV and LR performed a more detailed analysis of conditioning factors, but the results were uncertain. The result analyzed by GI may be more reliable but fluctuates with the amount of data.
4. The conventional algorithms are inferior to the advanced algorithms in accuracy and feature selection, but conventional algorithms have better resolvability and operability.

However, there are also some limitations of the present study:

1. There are possibilities for the combination of conventional and advanced algorithms, and further exploration is needed to improve prediction accuracy obviously.
2. Models need to be validated more reliably.

**Author Contributions:** Z.L., writing—original draft, methodology, and software; W.P., W.L. and H.H., review and validation; J.H. and K.L., reviewing and editing. K.J. and G.L. are responsible for the review and validation. All authors have read and agreed to the published version of the manuscript.

**Funding:** The completion of this work was supported by Guangzhou Collaborative Innovation Center of Natural Resources Planning and Marine Technology (No. 2023B04J0301), Key-Area Research and Development Program of Guangdong Province (No. 2020B0101130009), Guangdong Enterprise Key Laboratory for Urban Sensing, Monitoring and Early Warning (No. 2020B121202019), The Science and Technology Foundation of Guangzhou Urban Planning and Design Survey Research Institute (Grant No. RDI2220204128, RDI2220204031, RDI2220204037), and Postdoctoral Research Project of Guangzhou (20220402).

**Institutional Review Board Statement:** Not applicable.

**Informed Consent Statement:** Not applicable.

**Data Availability Statement:** Not applicable.

**Conflicts of Interest:** The authors declare no conflict of interest.

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
