# Peer review of "Exploration and Comparison of the Effect of Conventional and Advanced Modeling Algorithms on Landslide Susceptibility Prediction: A Case Study from Yadong Country, Tibet"

_applsci, doi:10.3390/app13127276_

Round 1

Author Response

Thank you for your letter and for the Reviewers’ comments concerning our manuscript entitled “Exploration and Comparison of the Effect of Conventional and Advanced Modeling Algorithms on Landslide Susceptibility Prediction: A Case Study from Yadong Country, Tibet” (ID: applies-2426759). Those comments are all valuable and very helpful for revising and improving our paper, as well as the important guiding significance to our research. We have studied the comments carefully and have made corrections which we hope meet with approval. Revised portions are marked in red on the paper. The main corrections in the paper and the response to the Reviewer’s comments are as flowing:

The manuscript entitled “Exploration and comparison of the effect of conventional and advanced modeling algorithms on Landslide susceptibility prediction: A case study from Yadong country, Tibet” that was submitted to “Applied Sciences-MDPI” explores the effect of advanced algorithm on the accuracy of landslide sucseptibility prediction results by comparing the performance of conventional and advanced algorithms through four representative models, namely, IV, LR, CatBoost, and CNN.The manuscript is of interest; however, the following comments are requested to be addressed by the authors:

Responds: Thank you for your approval of our work and we would revise the manuscript based on your comments and respond point by point.

  1. The English of the manuscript needs to be checked most preferably by a native English-speaking

person/professional service.

Responds: Yes, we have revised the manuscript by a native English-speaking person.

  1. The novelty & necessity of the manuscript should be presented and stressed in the “Introduction”

section.

Responds: Thank you and we have revised the Introduction on page 3, lines 57-59, 61-65.

  1. Please provide a more thorough list of references on the methods developed/applied on landslide

susceptibility analysis in the “Introduction” section by preferably supplying a table to demonstrate

the advantages and disadvantages of these methods. Towards the end of the “Introduction” section,

please mention the superiority & novelty of your work.

Responds: Thank you and we have revised the Introduction and mention the superiority & novelty of your work on page 4, lines 73-75.

  1. A relevant source of subjectivity and uncertainty is introduced when splitting the input parameters into an arbitrary number of classes with random break values. These choices affect the results. Please describe your solution.

Responds: Subjectivity and uncertainty are existent for LSM. Previous research derived performance measures for single-hold-out models by employing a single training set and an independent test set. In the study, 80% of the data sets were randomly chosen for training and 20% for validation in LSM and repeated the split 5 times. Finally, these models were evaluated using the average of 5 groups. We have discussed this point on page 12, lines 228-233, and page 19, lines 383-387.

  1. Please clearly state the main limitations of your method and findings.

Responds: Thank you and we have stated the main limitations of our manuscript on page 21, lines 412-415.

  1. What will be the consequence(s) if you use this algorithm/model for another region/Country?
    Responds: This comment is very interesting. The methods applied in the study are just a case for LSM and different algorithms have different emphases and generally, their performance varies with different study areas.

Once again, thank you very much for your comments and suggestions.

Reviewer 2 Report

Review: Exploration and comparison of the effect of conventional and advanced modeling algorithms on Landslide susceptibility prediction: A case study from Yadong country, Tibet

This paper presents a comparison between conventional and advanced methods for landslide susceptibility maps. Different criteria are used for the comparison and the results are adequately discussed in general, although some aspects require a deeper discussion.

The paper structure is correct and the sections are well organized in general. However, the paper must be fitted to the Applied Sciences and MDPI format, especially in the references list. Some figures should be revised and improved (indicated in the next sections). Tables should be also ordered and numbered correctly.

Title

Title seems to be correct.

Abstract

The abstract presents quite faithfully the methodology, main results and conclusions of the work, but the results seems to be incorrect (all of them are equal and too high) and do not correspond to those present in the results section.

Introduction

This section is correct in general. References are up-to-date and the main reviews are present.

Materials

Regarding the study area a better description of the geological context and a corresponding map (in figure 1) are necessary. 

Moreover, quality of figure 1 should be improved. Tibet Autonomous Region should be framed in China and Asia.

In data preparation, landslide inventory, I interpret that the final training and testing sample of landslides and non-landslides are points (as figure 1 shows). Are these points the centroids of landslide polygons? Please, clarify it.

In conditioning factors, first, the acronym should be indicated in those factors with acronyms (for instance maximum elevation difference, MED). Moreover, in line 112, authors could refer to MED, but the factor is not mentioned, and thus it seems that the description refers to slope.

A very important thing is that the author do not use lithology (only distance to faults) as a conditioning factor. Lithology is a crucial factor used by practically all the researchers of LSM. Its no inclusion here must be justified.   

Line 118. DEM has a resolution of 30 m, but what is the origin and the date of this model?

Lines 131 and the following. Again, what is the origin and date of the satellite image (Landsat?)?

Methods

Lines 148-189. The methodology has 5 steps. But either in figure 5, neither in the following text these 5 steps are clearly shown. Please, revise it.

In the IV expression (2), ln (Neperian) appears, but in the formula (3) log (decimal) does. Please revise it.

Line 180.  The text “… the and forward step …“ seems to be incorrect.

Description of CatBoost is something scarce. Although authors refer to another literature some details of the advantages and limitations of the algorithm should be provided. The same with CNN (Alexnet) structure (number of layers of convolution and pooling, dimension reduction in each one, etc.).

Table 2 is the first table numbered in the text. Please revise, the order of tables.

Results

Line 232. Table 2 in the text correspond to Table 3 (multicollinearity)

Line 236. Table 5 in the text correspond to Table 4 (model´s performance). Please, revise the numeration of tables.

Line 240-241. The sentence is a repetition of the above sentence.

Data between lines 236 and 243 correspond to training sample. Please, indicate clearly it. Moreover, accuracy and specially AUC are more global and complete indexes than sensitivity and specificity, so the evaluation should more based on them (discuss about it in the discussion section).

LSM classification, which scheme does it correspond to? Quantiles, natural breaks? And why dose the authors use this schema. It seems to be used a method that meet the criteria exposed below (lines 262-265); but for instance, why not use quantile methods to ensure a representation of all the susceptibility levels?

Line 272. Table 2 in the text seems to correspond to Table 5 (the first table 5).

The increasing value of IV along the susceptibility classes ensures the criteria 1 before mentioned. But other interesting criteria (if quantiles classification method is not used) is a the decreasing area of susceptibility classes (large extension for lower susceptibility and wide extension for higher susceptibility), thus ensuring the elaboration of not so much conservative maps. In this case, the susceptibility maps are too conservative, since the very high susceptibility has a large extension, especially in the IV model. What can you say about this?

Lines 280-284. This paragraph could correspond more to a discussion section than a result description.  

Subsection 4.3 could be before the subsection 4.2 in a more logical order.

Regarding the factor analysis. Elevation has a chaotic distribution of landslides. Slope and MED presents a logic (increasing) distribution. However plane curvature and profile curvature do not show a logical distribution; thus, landslides concentrate in plane curvature near 0 and in convex sections (positive values of profile curvature), while usually landslide concentrate in both negative values (concave forms). Moreover, regarding TWI, the same occurs, since landslides concentrate at lower values, while usually concentrate in higher values. Can you explain it?

Menwhile, distance to faults, streams and roads show a logical distribution. But NDVI and precipitation, also show a chaotic distribution that should be discussed better in the discussion section.

The negative coefficient of DTR in regression equation can be explained (probably in discussion section) by the increasing density of landslides with the distance. And why you use only 5 factor in logistic regression? Please, justify it.

Table 3 seems to be the (second) Table 5 in table list. Please, again, revise this numeration. Moreover, only 6 factors are shown. Does it mean that you use only 6 factors in the advanced methods, or you use the 11 factors (and reduce the dimensionless with CNN methods? Please clarify it, for the different methods.

Discussion

In general, authors show interesting observation by comparing methods. This section sounds well (feature selection by CNNs, importance of factor analysis and identifying the major conditions of landsides, etc.). However, along the previous section, I have mentioned some aspects that should be included in this section (AUC values as the most important validation index, role of some factors not so discussed, etc.).

Other aspect (lines 344-345) is the difficulty for non-experts regarding to tune the parameters of ML methods. How can this challenge be overcome?

Lines 364-365. Here authors refer to LSM classification methods, but it is not clear what method is used.

Conclusions

This section seems to be correct as the abstract.

References

The reference list must be revised in order to fit it to MDPI style.

Author Response

Thank you for your letter and for the Reviewers’ comments concerning our manuscript entitled “Exploration and Comparison of the Effect of Conventional and Advanced Modeling Algorithms on Landslide Susceptibility Prediction: A Case Study from Yadong Country, Tibet” (ID: applies-2426759). Those comments are all valuable and very helpful for revising and improving our paper, as well as the important guiding significance to our research. We have studied the comments carefully and have made corrections which we hope meet with approval. Revised portions are marked in red on the paper. The main corrections in the paper and the response to the Reviewer’s comments are as flowing:

This paper presents a comparison between conventional and advanced methods for landslide susceptibility maps. Different criteria are used for the comparison and the results are adequately discussed in general, although some aspects require a deeper discussion.

The paper structure is correct and the sections are well organized in general. However, the paper must be fitted to the Applied Sciences and MDPI format, especially in the references list. Some figures should be revised and improved (indicated in the next sections). Tables should be also ordered and numbered correctly.

Responds: Thank you for your approval of our work and we will revise the manuscript based on your comment and respond one by one.

Title

Title seems to be correct.

Responds: Thank you.

Abstract

The abstract presents quite faithfully the methodology, main results and conclusions of the work, but the results seems to be incorrect (all of them are equal and too high) and do not correspond to those present in the results section.

Responds: We are sorry for the mistakes and we have revised the results on page 1, lines 26-30.

Introduction

This section is correct in general. References are up-to-date and the main reviews are present.

Responds: Thank you.

Materials

Regarding the study area a better description of the geological context and a corresponding map (in figure 1) are necessary. 

Responds: We have revised the description of study area on page 5, lines 81-96 and also revised figure 1.

Moreover, quality of figure 1 should be improved. Tibet Autonomous Region should be framed in China and Asia.

Responds: We have revised Figure 1.

In data preparation, landslide inventory, I interpret that the final training and testing sample of landslides and non-landslides are points (as figure 1 shows). Are these points the centroids of landslide polygons? Please, clarify it.

Responds: Yes, the landslide locations are the centroids of landslide polygons and we have clarified it on page 6, lines 107-112.

In conditioning factors, first, the acronym should be indicated in those factors with acronyms (for instance maximum elevation difference, MED). Moreover, in line 112, authors could refer to MED, but the factor is not mentioned, and thus it seems that the description refers to slope.

Responds: We have revised it on page 6, lines 118-119.

A very important thing is that the author do not use lithology (only distance to faults) as a conditioning factor. Lithology is a crucial factor used by practically all the researchers of LSM. Its no inclusion here must be justified.   

Responds: Thank you for your suggestion. The geological composition of the study area is primarily composed of shale, limestone, and dolomite. The lithology distribution in the study area is simple. Thus, we select distance to faults as a more important factors.

Line 118. DEM has a resolution of 30 m, but what is the origin and the date of this model?

Responds: We have added related information on page 7, lines 130-131.

Lines 131 and the following. Again, what is the origin and date of the satellite image (Landsat?)?

Responds: We have added the related information on page 6, lines 136-137 and page 7, lines 151-152.

Methods

Lines 148-189. The methodology has 5 steps. But either in figure 5, neither in the following text these 5 steps are clearly shown. Please, revise it.

Responds: We are sorry for the mistakes. It should be 4 steps and we have revised the Figure 5.

In the IV expression (2), ln (Neperian) appears, but in the formula (3) log (decimal) does. Please revise it.

Responds: Thank you and we have revised formula 3, on page 9, line 181.

Line 180.  The text “… the and forward step …“ seems to be incorrect.

Responds: We have revised this sentence on page 10, lines 194-195.

Description of CatBoost is something scarce. Although authors refer to another literature some details of the advantages and limitations of the algorithm should be provided. The same with CNN (Alexnet) structure (number of layers of convolution and pooling, dimension reduction in each one, etc.).

Responds: We have added related information about CatBoost on page 10 , lines 204-205, page 11, line 216-217.

Table 2 is the first table numbered in the text. Please revise, the order of tables.

Responds: We have numbered the tables.

Results

Line 232. Table 2 in the text correspond to Table 3 (multicollinearity)

Responds: Thank you, and we have revised the number.

Line 236. Table 5 in the text correspond to Table 4 (model´s performance). Please, revise the numeration of tables.

Responds: Thank you, and we have revised the number.

Line 240-241. The sentence is a repetition of the above sentence.

Responds: We have deleted the repetition.

Data between lines 236 and 243 correspond to training sample. Please, indicate clearly it. Moreover, accuracy and specially AUC are more global and complete indexes than sensitivity and specificity, so the evaluation should more based on them (discuss about it in the discussion section).

Responds: We have clearly it on line 252-253.

LSM classification, which scheme does it correspond to? Quantiles, natural breaks? And why dose the authors use this schema. It seems to be used a method that meet the criteria exposed below (lines 262-265); but for instance, why not use quantile methods to ensure a representation of all the susceptibility levels?

Responds: The natural breaks method was applied to reclassify landslide susceptibility into five classes: very low, low, moderate, high, and very high. Page 12, lines 243-244.

Line 272. Table 2 in the text seems to correspond to Table 5 (the first table 5).

Responds: Yes, we have revised the number.

The increasing value of IV along the susceptibility classes ensures the criteria 1 before mentioned. But other interesting criteria (if quantiles classification method is not used) is a the decreasing area of susceptibility classes (large extension for lower susceptibility and wide extension for higher susceptibility), thus ensuring the elaboration of not so much conservative maps. In this case, the susceptibility maps are too conservative, since the very high susceptibility has a large extension, especially in the IV model. What can you say about this?

Responds: Your comment is very interesting and meaningful. We have also discussed this point in the Discussion. If we applied the equal intervals method to reclassify the landslide susceptibility, the results of IV would be different. Besides, the division of the conditioning factors interval is also influential.

Lines 280-284. This paragraph could correspond more to a discussion section than a result description.  

Responds: It is true and we have adjusted the order, on page 18, lines 348-352.

Subsection 4.3 could be before the subsection 4.2 in a more logical order.

Responds: We have adjusted the order.

Regarding the factor analysis. Elevation has a chaotic distribution of landslides. Slope and MED presents a logic (increasing) distribution. However plane curvature and profile curvature do not show a logical distribution; thus, landslides concentrate in plane curvature near 0 and in convex sections (positive values of profile curvature), while usually landslide concentrate in both negative values (concave forms). Moreover, regarding TWI, the same occurs, since landslides concentrate at lower values, while usually concentrate in higher values. Can you explain it?

Responds: Your comment is very interesting and meaningful. As we discuss that the bivariate methods like IV, FR, and CF, the performance is mainly up to the division of the conditioning factors interval. However, there is no consensus on the size or number of intervals although some methods such as natural breaks and equal intervals have been applied.

Menwhile, distance to faults, streams and roads show a logical distribution. But NDVI and precipitation, also show a chaotic distribution that should be discussed better in the discussion section.

Responds: We have added related information on discussion, page 19, lines 382-385.

The negative coefficient of DTR in regression equation can be explained (probably in discussion section) by the increasing density of landslides with the distance. And why you use only 5 factor in logistic regression? Please, justify it.

Responds: In the current study, the LR model was performed in SPSS software, the forward stepwise method was adopted to screen valuation indexes. The conditioning factors were calculated as independent variables, whereas dependent variables represent the occurrence of landslides. During the last step of the analysis, all variables were significant at less than 0.05, so no additional variables were included, on page 10, lines 194-198.

Table 3 seems to be the (second) Table 5 in table list. Please, again, revise this numeration. Moreover, only 6 factors are shown. Does it mean that you use only 6 factors in the advanced methods, or you use the 11 factors (and reduce the dimensionless with CNN methods? Please clarify it, for the different methods.

Responds: Thank you and we have checked all the Table list. 11 factors were involved in the modelling for LR, CatBoost, IV, and CNN. The forward stepwise method was adopted to screen valuation indexes for LR, so only 5 factors in logistic regression. Gini importance (GI) is defined as the total reduction in average nodal impurities across all trees. GI was also applied to analyze the relative importance of different factors, exploring the factor’s contribution to landslide occurring. Table 5 showed the rank of the main conditioning factors of this study.

Discussion

In general, authors show interesting observation by comparing methods. This section sounds well (feature selection by CNNs, importance of factor analysis and identifying the major conditions of landsides, etc.). However, along the previous section, I have mentioned some aspects that should be included in this section (AUC values as the most important validation index, role of some factors not so discussed, etc.).

Responds: It is true that authors show interesting observations by comparing methods. We have adjust the comparison of these models on page 18, lines 348-352, line 356-357.

Other aspect (lines 344-345) is the difficulty for non-experts regarding to tune the parameters of ML methods. How can this challenge be overcome?

Responds: This comment is interesting. Some algorithms for optimizing parameters have been applied, but what we want to explain is that the realization of the machine algorithm is more difficult than the conventional statistical, and the threshold is higher.

Lines 364-365. Here authors refer to LSM classification methods, but it is not clear what method is used.

Responds: Natural breaks and equal intervals on page 19, lines 384-385.

Conclusions

This section seems to be correct as the abstract.

Responds: Thank you.

References

The reference list must be revised in order to fit it to MDPI style.

Responds: We have revised the reference list.

Reviewer 3 Report

The present study entails a framework to compare the performance of two conventional [information value (IV) and logistic regression (LR)] and two advanced [categorical boosting (CatBoost) and conventional neural networks (CNN)] algorithms for Yadong county, Tibet. The paper is generally well-written, but the following corrections and suggestions should be incorporated before publication:

Major Revision

1.      The abstract needs to be better structured, clearly focussing on the importance of the present study.

2.      Line 17-18: Modeling as an essential step, various algorithms have been applied to LSM. This line doesn’t convey anything.

3.      Introduction lacks a proper flow of sentences, and English is very poor. Many sentences need to be rephrased.

4.      The introduction section needs to address the key research gaps in the present studies by highlighting the poor or inadequate performance of the conventional models. The authors also need to consider the application of the latest artificial intelligence-based models which are in practice.

5.      The study area description is very poorly written. Relatively high geological disasters are in the area; which disasters? Any details or data to support the statement? MMI scale for seismicity? Legends of Figure 1 have different font size and style and what is DEM in legend?

6.      Which data is used for the preparation of the landslide inventory?

7.      How the 496 landslides were identified and from where (aerial photograph, literature or through fieldwork?)?

8.      992 samples of what size? Is it a single pixel or a minimum area?

9.      Figures are not clear.

10.  The methodology section should include the procedure for landslide data split into training and testing datasets and why the 80-20 ratio was chosen. Also, how many times the dataset is split (k-fold cross-validation) while calibrating the models should be highlighted.

11.  The discussion needs a critical analysis of the results and how these models compare to similar studies in similar regions. Also, the authors need to clearly elaborate on why some landslide causative factors have more influence on landslide occurrence in the region as compared to others. 

The quality of English is very poor. 

Author Response

Thank you for your letter and for the Reviewers’ comments concerning our manuscript entitled “Exploration and Comparison of the Effect of Conventional and Advanced Modeling Algorithms on Landslide Susceptibility Prediction: A Case Study from Yadong Country, Tibet” (ID: applies-2426759). Those comments are all valuable and very helpful for revising and improving our paper, as well as the important guiding significance to our research. We have studied the comments carefully and have made corrections which we hope meet with approval. Revised portions are marked in red on the paper. The main corrections in the paper and the response to the Reviewer’s comments are as flowing:

The present study entails a framework to compare the performance of two conventional [information value (IV) and logistic regression (LR)] and two advanced [categorical boosting (CatBoost) and conventional neural networks (CNN)] algorithms for Yadong county, Tibet. The paper is generally well-written, but the following corrections and suggestions should be incorporated before publication:

Responds: Thank you for your approval of our work and we would revise the manuscript based on your comments and respond point by point.

  1. The abstract needs to be better structured, clearly focussing on the importance of the present study.

Responds: We have revised the abstract and emphasized the significance and necessity of this paper.

  1. Line 17-18: Modeling as an essential step, various algorithms have been applied to LSM. This line doesn’t convey anything.

Responds: We have revised this sentence on page 1, lines 17-18.

  1. Introduction lacks a proper flow of sentences, and English is very poor. Many sentences need to be rephrased.

Responds: We have revised the introduction and improved the quality of English.

  1. The introduction section needs to address the key research gaps in the present studies by highlighting the poor or inadequate performance of the conventional models. The authors also need to consider the application of the latest artificial intelligence-based models which are in practice.

Responds: Thank you, and we have highlighted the poor or inadequate performance of the conventional models on page 3, lines 57-59, 61-65.

  1. The study area description is very poorly written. Relatively high geological disasters are in the area; which disasters? Any details or data to support the statement? MMI scale for seismicity? Legends of Figure 1 have different font size and style and what is DEM in legend?

Responds: We have revised the description of the study area on page 4-5. We have revised Figure 1.

  1. Which data is used for the preparation of the landslide inventory?

Responds: Remote sensing images, literature, and historical reports, and conducting extensive fieldwork. On page 6, lines 107-108.

  1. How the 496 landslides were identified and from where (aerial photograph, literature or through fieldwork?)?

Responds: Remote sensing images, literature, and historical reports, and conducting extensive fieldwork. On page 6, lines 107-112.

  1. 992 samples of what size? Is it a single pixel or a minimum area?

Responds: Landslides are mainly shallow landslides with volumes ranging from 100m³ to 1000m³. The landslide samples are bounded by polygons containing the entire perimeter, page 6, line 107-112.

  1. Figures are not clear.

Responds: We have improved the resolution of the Figures.

  1. The methodology section should include the procedure for landslide data split into training and testing datasets and why the 80-20 ratio was chosen. Also, how many times the dataset is split (k-fold cross-validation) while calibrating the models should be highlighted.

Responds: We have added the relevant notes on page 12, lines 228-233.

  1. The discussion needs a critical analysis of the results and how these models compare to similar studies in similar regions. Also, the authors need to clearly elaborate on why some landslide causative factors have more influence on landslide occurrence in the region as compared to others. 

Responds: Thank you and we have revised the discussion based on your comment, on pages 17-19.

Round 2

Reviewer 2 Report

Authors have attended most suggestions I made in the first revision. However, some minor points in the text, but also in the Figures, Tables and references. Then, my overall recommendations is minor revision before publication.

Regarding the text, some aspects have not been sufficiently addressed:

·        The description of geological setting, especially of the southern region, continues being too simply.

·        Not using of lithology as determinant factor should be discussed in the manuscript not only in the response.

·        The explanation to my suggestion “Table 3 seems to be the (second) Table 5 in table list. Please, again, revise this numeration. Moreover, only 6 factors are shown. Does it mean that you use only 6 factors in the advanced methods, or you use the 11 factors (and reduce the dimensionless with CNN methods?” should be included in the manuscript.

Regarding the figures, Figure 1 is not modified in the revised manuscript as you say in the response. Remember my suggestions:

·        A better description of the geological context and a corresponding map (in figure 1) are necessary. 

·        Moreover, quality of figure 1 should be improved. Tibet Autonomous Region should be framed in China and Asia.

Table 1 is not cited in the text and Table 5 is repeated in the tables description, so Table 6 does not appear.

References are still not fitted to RS usage. Please revise them for publication.

Author Response

Dear Editors and Reviewers

Thank you for your letter and for the Reviewers’ comments concerning our manuscript entitled “Exploration and Comparison of the Effect of Conventional and Advanced Modeling Algorithms on Landslide Susceptibility Prediction: A Case Study from Yadong Country, Tibet” (ID: applies-2426759). Those comments are all valuable and very helpful for revising and improving our paper, as well as the important guiding significance to our research. We have studied the comments carefully and have made corrections which we hope meet with approval. Revised portions are marked in red on the paper. The main corrections in the paper and the response to the Reviewer’s comments are as flowing:

Authors have attended most suggestions I made in the first revision. However, some minor points in the text, but also in the Figures, Tables and references. Then, my overall recommendations is minor revision before publication.

Responds: Thank you and we would further revise the paper.

Regarding the text, some aspects have not been sufficiently addressed:

  1. The description of geological setting, especially of the southern region, continues being too simply.

Responds: We have added the related information on page 5, lines 94-97.

  • 2.Not using of lithology as determinant factor should be discussed in the manuscript not only in the response.

Responds: Yes, we have added the information on page 20, lines 398-401.

  • 3.The explanation to my suggestion “Table 3 seems to be the (second) Table 5 in table list. Please, again, revise this numeration. Moreover, only 6 factors are shown. Does it mean that you use only 6 factors in the advanced methods, or you use the 11 factors (and reduce the dimensionless with CNN methods?” should be included in the manuscript.

Responds: We have revised the numbers of Tables.

4.Regarding the figures, Figure 1 is not modified in the revised manuscript as you say in the response. Remember my suggestions

Responds: We have revised Figure 1, on page 33, line594.

  • 4. A better description of the geological context and a corresponding map (in figure 1) are necessary. 

Responds: We have improved the quality of Figure.1 and added the related information on page 5, lines 94-97.

  • 5.Moreover, quality of figure 1 should be improved. Tibet Autonomous Region should be framed in China and Asia.

Responds: We have revised Figure 1.

        6.Table 1 is not cited in the text and Table 5 is repeated in the tables   description, so Table 6 does not appear.

Responds: Table 1 is cited on page 14, lines 274. Table 5 is repeated in the tables description, so Table 6 does not appear. We have replaced Table 5 with Table 6.

  1. References are still not fitted to RS usage. Please revise them for publication.

Responds: We have revised them.

Reviewer 3 Report

The authors have revised the manuscript taking into consideration most of the suggestions, which has significantly improved the manuscript.  

Moderate English checks required.

Author Response

Dear Editors and Reviewers

Thank you for your letter and for the Reviewers’ comments concerning our manuscript entitled “Exploration and Comparison of the Effect of Conventional and Advanced Modeling Algorithms on Landslide Susceptibility Prediction: A Case Study from Yadong Country, Tibet” (ID: applies-2426759). Those comments are all valuable and very helpful for revising and improving our paper, as well as the important guiding significance to our research. We have studied the comments carefully and have made corrections which we hope meet with approval. Revised portions are marked in red on the paper. The main corrections in the paper and the response to the Reviewer’s comments are as flowing:

The authors have revised the manuscript taking into consideration most of the suggestions, which has significantly improved the manuscript.  

Responds: Thank you for your approval and kindly work on our manuscript.

Moderate English checks required.

Responds: We have checked the English again.